# Preanalytical Impact of Incomplete K_2_EDTA Blood Tube Filling in Molecular Biology Testing

**DOI:** 10.3390/diagnostics14171934

**Published:** 2024-09-02

**Authors:** Marco Benati, Laura Pighi, Elisa Paviati, Sara Visconti, Giuseppe Lippi, Gian Luca Salvagno

**Affiliations:** Section of Clinical Biochemistry, Department of Engineering for Innovative Medicine (DIMI), University of Verona, Piazzale L.A. Scuro 10, 37134 Verona, Italy; laura.pighi2@aovr.veneto.it (L.P.); paviati.elisa@univr.it (E.P.); sara.visconti@aovr.veneto.it (S.V.); giuseppe.lippi@univr.it (G.L.); gianluca.salvagno@univr.it (G.L.S.)

**Keywords:** quality, molecular biology, preanalytical variability, DNA, RNA

## Abstract

Background and aims: The aim of this study was to investigate the possible preanalytical effect of incomplete filling of blood tubes on molecular biology assays. Materials and methods: The study population consisted of 13 healthy volunteers from whom 11 mL of whole blood was collected and then distributed in different volumes (1.5, 3.0, and 6.0 mL, respectively) into three 6.0 mL spray-dried and evacuated K_2_EDTA blood tubes. Automated RNA extraction was performed using the Maxwell^®^ CSC RNA Blood Kit. DNA was extracted with a MagCorePlusII, with concomitant measurement of glyceralde-hyde-3-phosphate dehydrogenase (GAPDH) gene expression. The nucleic acid concentration was calculated using the NanoDrop 1000 spectrophotometer, and purity was assessed using A260/280 and A260/230 absorbance ratios. Results: The RNA concentration was higher in the tubes filled with 1.5 and 3.0 mL of blood than in the reference 6 mL filled tube. The RNA 260/280 and RNA 260/230 ratios did not differ significantly between the differently filled blood tubes. The DNA concentration remained constant in the differently filled tubes. Compared to the 6.0 mL reference filled tube, the 1.5 mL and 3.0 mL filled blood tubes displayed a lower DNA 260/280 nm ratio. The DNA 260/230 ratio did not differ significantly in any of the variously filled tubes. Compared to the 6.0 mL reference filled blood tube, the 1.5 mL and 3.0 mL filled blood tubes showed a significant increase in the GAPDHcycle threshold. Conclusions: Our results suggest that underfilling of K_2_EDTA blood tubes may be a modest but analytically significant source of bias in molecular biology testing.

## 1. Introduction

Preanalytical variability plays a crucial role throughout the total testing process [1]. It is now widely recognized that preanalytical mistakes (i.e., involving all manual activities from test requests to sample preparation for testing) are the main source of problems in clinical laboratories [2]. Among the various potential sources of extra-analytical problems, the collection of an insufficient volume of a sample is a rather frequent occurrence (e.g., about 10% of all preanalytical mistakes) and poses a serious threat to the quality of tests for at least two main reasons. The first aspect is quite intuitive and is directly related to the total volume of a sample. Although the modern generation of laboratory analyzers requires only a few microliters for performing most tests, some results must be suppressed if the total volume of a sample needed to perform all required analyses exceeds the available material [3].

Most laboratory analyzers are now equipped with volumetric sensors that should avoid analytical errors due to aspiration of insufficient sample volume for testing. However, additional aspects need to be considered, such as the high risk of evaporation, as smaller volumes have a higher surface area to total volume ratio but also, and perhaps most importantly, an impaired blood-to-additive ratio [3]. For certain analyses, with hemostasis testing being a paradigmatic example, there is a fixed blood-to-additive ratio that should always be fulfilled. Underfilling the blood coagulation tube results in a change in the normal blood-to-additive ratio (i.e., typically 1:10 with buffered sodium citrate), thus leaving a higher concentration of anticoagulant in the sample, which would result in false prolongation of various hemostasis tests (the activated partial thromboplastin time [APTT]-derived tests are most sensitive to blood tube underfilling) [4]. Underfilling the blood tube and the resulting change in the blood-to-additive ratio may also have an impact on other laboratory tests, especially those involving particular enzymes (creatine kinase, gamma-glutamyl transferase, lactate dehydrogenase) [5,6], blood gas analysis [7] and HbA1c [8], among others. However, in the absence of information on the effects of underfilling blood tubes on molecular biology assays, we planned a specific series of experiments, based on the best of our knowledge, to investigate whether a lower volume of blood collected in evacuated blood tubes may impair the quality of DNA and RNA extraction.

## 2. Materials and Methods

### 2.1. Blood Samples

The study population consisted of 13 healthy volunteers (4 males and 9 females; with a mean age of 47.3 ± 11–6 years and ranging from 26–66 years) who were recruited from the staff of the local laboratory medicine service at the University Hospital of Verona (Italy). All volunteers provided informed consent for participating in this study, which was performed in accordance with the Declaration of Helsinki. The study was cleared by the Ethical Committee of the University Hospital of Verona (970CESC; 20 July 2016).

A total volume of 11 mL of whole blood was drawn from a peripheral vein of the arm by direct venipuncture using a 21-gauge disposable needle (KDL, Nanchang, Jiangxi, China) connected to a standard 10 mL syringe (Plastipak Luer-Lok 10 mL Syringe, Becton Dickinson, Madrid, Spain). This was then distributed with different volumes into three 6 mL spray-dried 10.8 mg K_2_EDTA evacuated blood tubes (Vacutest, Kima, Arzergrande, Padova, Italy) as follows: 1.5 mL into the first blood tube, 3 mL into the second blood tube, and 6 mL into the last blood tube, respectively. Blood tube filling was completed as soon as possible, thus preventing spurious clotting of blood within the syringe. Extraction and polymerase chain reaction (PCR) analysis of RNA and DNA were then performed.

### 2.2. RNA Extraction and RT-PCR

Automated RNA extraction was performed using 1 mL of blood from all of the tubes, using the Maxwell^®^ CSC RNA Blood Kit (Maxwell instrument, Promega, Walldorf, Germany) according to the manufacturer’s specifications. Nucleic acid concentration was determined by measuring the absorbance of ultraviolet light on the NanoDrop-1000 spectrophotometer (Thermo Fisher Scientific, Waltham, MA, USA). Additionally, as an indicator of sample purity, the ratios of absorbance values of 260 nm vs. 280 nm (A260/A280) and 260 nm vs. 230 nm (A260/A230) were determined.

To check the quality and quantity of extracted RNA, real-time PCR was performed. Specifically, 100 ng of extracted RNA was retrotranscribed using the M-MLV (Moloney Murine Leukemia Virus) Reverse Transcriptase (RT) (Thermo Fisher Scientific, Waltham, MA, USA) with random hexamers, according to the manufacturer’s instructions. To determine if the purified RNA samples could be efficiently reverse-transcribed and amplified, the samples were assayed using quantitative PCR methods without post-PCR processing. We adopted an RT-PCR protocol to detect the quantitative Abelson murine leukemia (ABL) transcript. The primer sequences (Metabion International AG, Planegg, Germany) used and the expected transcript amplicons generated are summarized in Table 1. TaqMan real-time quantitative PCR amplification reactions were carried out in an AB 7500 fast Real-Time system (Thermo Fisher Scientific, Waltham, MA, USA). Briefly, a 20 µL reaction mixture consisted of 10 µL of TaqMan Universal PCR Master Mix, 200 nM of primers, and 100 nM TaqMan probe. The following thermal conditions were used: activation at 95 °C for 10 min, cyclic denaturation at 95 °C for 35 s, and cyclic annealing and extension at 60 °C for 1 min. A calibration curve with common plasmid solutions of ABL produced independently from the stock solutions of ERMAD623 (Sigma-Aldrich, Oakville, ON, Canada) was included in every PCR run.

### 2.3. DNA Extraction and Real-Time PCR

DNA was extracted from each of the blood samples using a MagCorePlusII (RBC Bioscience Corp., New Taipei City, Taiwan) in accordance with the manufacturer’s instructions. DNA concentration (expressed as ng/μL) was calculated directly using a NanoDrop-1000 spectrophotometer (Thermo Fisher Scientific, Waltham, MA, USA), and DNA purity for assessing protein and salt contaminants was based on the A260/280 and A260/230 absorbance ratios.

A concomitant measurement of the glyceraldehyde-3-phosphate dehydrogenase (GAPDH) gene was used to check for experimental variations in the amount and quality of DNA in each tube. The real-time PCR was performed using an AB 7500 fast Real-Time system (Thermo Fisher Scientific, Waltham, MA, USA) according to the manufacturer’s instructions, with the Sybr Green Master Mix kit (Thermo Fisher Scientific, Waltham, MA, USA). The GAPDH primer sequences (Metabion International AG, Martinsried, Germany) that were used are summarized in Table 1. Ct (cycle threshold) values were obtained for each sample.

### 2.4. Statistical Analysis

The statistical analysis was performed using GraphPad Prism (Version 5.0; GraphPad Software, San Diego, CA, USA) and was based on a paired Wilcoxon signed-rank test for establishing the significance of differences from the reference (6 mL filled) specimen. Statistical significance was set at *p* < 0.05. The results from the different filling tubes in this study, including RNA and DNA concentration, 260/280 ratio, 269/230 ratio, and Ct (cycle threshold) were reported as mean and standard deviation (SD).

## 3. Results

### 3.1. RNA Analysis

The results of this series of experiments are shown in Figure 1. The RNA concentration was found to be significantly higher in the 1.5 mL filled blood tube compared to the reference 6 mL filled blood tube (mean 62.44 ± 21.36 versus 56.58 ± 19.94 ng/µL; *p* = 0.001). A marginally significant difference was also found when the mean RNA concentration was compared between the 3 mL filled blood tube and the reference 6 mL filled blood tube (59.84 ± 20.75 versus 56.58 ± 19.94 ng/µL; *p* = 0.021). The differences in concentrations were generally modest, indicating that the concentration of nucleic acid may be minimally affected by tube filling. No differences in RNA concentration were observed between the 1.5 mL and 3 mL filled samples (mean 62.44 ± 21.36 versus 59.84 ± 20.75 ng/µL, *p* = 0.414). No significant differences were observed for the 260/280 and 260/230 ratios among all differently filled blood tubes (*p* > 0.05 for all).

The ABL expression levels displayed analytically significant increases when comparing the 1.5 filled blood tube with the reference 6 mL blood tube (mean 8225 ± 1950 versus 7343 ± 1731 copy number; *p* = 0.026), but no differences were observed between the 3 mL and 6 mL filled blood tubes (mean 7460 ± 1794 versus 7343 ± 1731 copy number, *p* = 0.735). Additionally, ABL expression levels did not significantly differ between the 1.5 mL and 3 mL filled blood tubes (*p* = 0.068).

### 3.2. DNA Analysis

The results of this series of experiments are shown in Figure 2, revealing a wide spectrum of DNA concentrations in all blood tubes (between 54.6 and 102.94 ng/µL, with a mean value of 77.48 ng/µL, without displaying statistically significant differences (all *p* > 0.05). The ratio between the absorbance at 260 and 280 nm was found to be marginally but significantly lower in 1.5 mL filled blood tubes than in the reference 6 mL filled blood tubes (mean 1.75 ± 0.05 versus 1.77 ± 0.04; *p* = 0.047). The ratio was also marginally but significantly lower when comparing the 3 mL filled blood tubes with the 6 mL filled blood tubes (1.75 ± 0.05 versus 1.77 ± 0.04; *p* = 0.021). No differences were observed when comparing the 1.5 mL and the 3 mL filled blood tubes (1.75 ± 0.05 versus 1.75 ± 0.05; *p* = 0.552). No significant differences were observed for the 260/230 ratio among all differently filled blood tubes (all *p* > 0.05).

Importantly, the Ct value of GAPDH amplification displayed analytically significant increases in the blood from the 1.5 mL and the 3 mL filled blood tubes compared to the reference 6 mL filled blood tube (*p* = 0.026 and 0.017, respectively). No differences were found in the Ct value of GAPDH between the 1.5 mL and 3 mL filled blood tubes (*p* = 0.339).

## 4. Discussion

The extraction of DNA and RNA represents one of the most critical methods in molecular biology, as the quality and integrity of the nucleic acid will have a direct impact on the results of the subsequent testing [9]. Some previous reports showed that errors and pitfalls in the preanalytical phase of molecular biology testing may have a substantial impact on test results and, consequently, on clinical decision making. For example, collection techniques, sample stability, and extraction kits were found to have a substantial impact on the quality of DNA specimens before downstream molecular analyses [10]. Historically, EDTA has been recommended as the anticoagulant of choice for hematological testing because it allows the best preservation of cellular components and morphology of blood cells since it has an inhibitory effect on the nuclease present in blood [11,12]. In our study, we found that RNA concentrations began to decrease depending on the filling level of the blood tube, while DNA concentrations remained unchanged. The fluctuations in RNA concentration could be due to the different activity of RNases in the presence of a high EDTA concentration. A study by Grunberg et al. [13] showed that EDTA dramatically reduces RNase I activity by capturing ionized calcium. Brännvall et al. [14] also showed that ionized magnesium acts in the RNase P RNA-catalyzed reaction to permit appropriate folding of RNA, thus facilitating the interaction with its RNA substrate and promoting efficient and correct cleavage of newly released RNA, whereas DNA is almost insensitive to this reaction.

However, since ribose residues carry hydroxyl groups in both the 2′ and 3′ positions, RNA is chemically more reactive than DNA and more susceptible to cleavage by contaminating RNases. This would ultimately explain why differences in RNA concentrations exist between different filling tubes but not in DNA concentrations. In addition, physiological concentrations of chelated ionized magnesium promote the thermostability of RNA [15]. The expression differences measured with the real-time PCR are similar to those obtained from RNA or DNA, confirming that PCR efficiency is generally unaffected by EDTA, but the acid nucleic quality was higher in the 1.5 mL than in the 6 mL filled blood tubes for RNA. However, the Ct value of GAPDH was higher in the 1.5 mL than in the 6 mL filled blood tubes, in agreement with the study by Cai et al. [16], which showed that excessive EDTA can become a PCR inhibitor through binding of DNA polymerase co-factors (magnesium ions), which could have an impact on DNA amplification. Further studies are also needed to elucidate the role of EDTA in PCR performance.

One of the most commonly used methods to estimate nucleic acid concentration is the measurement of sample absorbance at 260 nm. The ratio of absorbance between 260 and 280 nm is then used to assess the nucleic acid average concentration in a mixture, along with its purity [17,18]. A ratio of ~1.8 is generally accepted as “pure” for nucleic acids [19]. If the ratio is appreciably lower (≤1.6), it may indicate the presence of proteins, phenol, or other contaminants that absorb strongly at or near 280 nm. The 260/230 ratio is used as a secondary measure of nucleic acid purity [20]. Expected 260/230 values for “pure” nucleic acid are commonly within the range of 2.0 and 2.2.

In our study, we found no significant differences in the RNA ratio between 260/280 and 260/230, whilst significantly different 260/280 ratios were found for DNA in different filled blood tubes, which could be attributable to some factors previously mentioned, in particular protein contamination. In the study of Janeckiet al. [21], the authors found that EDTA destabilizes metalloprotein structures, and this could influence the 280 absorbances. Moreover, this result could also be due to the modest variations in the pH or ionic strength of the spectrophotometric solution affecting the A260/280 ratios, as earlier demonstrated by Wilfinger et al. [22].

The techniques used for whole blood preservation and storage, prior to processing into plasma and DNA extraction, will have a growing impact on the quality of analysis as the number of molecular analyses for investigating mutation profiles increases for both research and clinical purposes. It is obvious that reducing sample deterioration while storing whole blood will become more crucial than ever. Given the lack of research on the comparability of blood volume percentage in EDTA-filled blood tubes for molecular biology assay measurement, our findings indicate that variability in molecular biology testing could also be caused by inappropriate drawing. Although this study confirms the importance of the preanalytical phase and the correct filling of blood tubes for molecular testing, it also shows that samples do not have to be rejected when the blood tube is underfilled by 75% or less. This is particularly important when drawing blood from patients with difficult veins, neonates, the elderly, and critically ill patients, where iatrogenic anemia should be prevented. Nevertheless, we believe that adequate filling of the tubes is still necessary to obtain high-quality and especially reproducible molecular biology test results.

Our study has some limitations, with the first being the sample size. Further studies will be needed to explore the impact of EDTA concentration when assessing nucleic acids in larger populations. We also acknowledge that the use of capillary electrophoresis, which was not available at our institution, would allow for better accuracy in quantifying nucleic acids and establishing nucleic acid integrity by analyzing electropherograms and calculating the RNA integrity number (RIN).

## Figures and Tables

**Figure 1 diagnostics-14-01934-f001:**
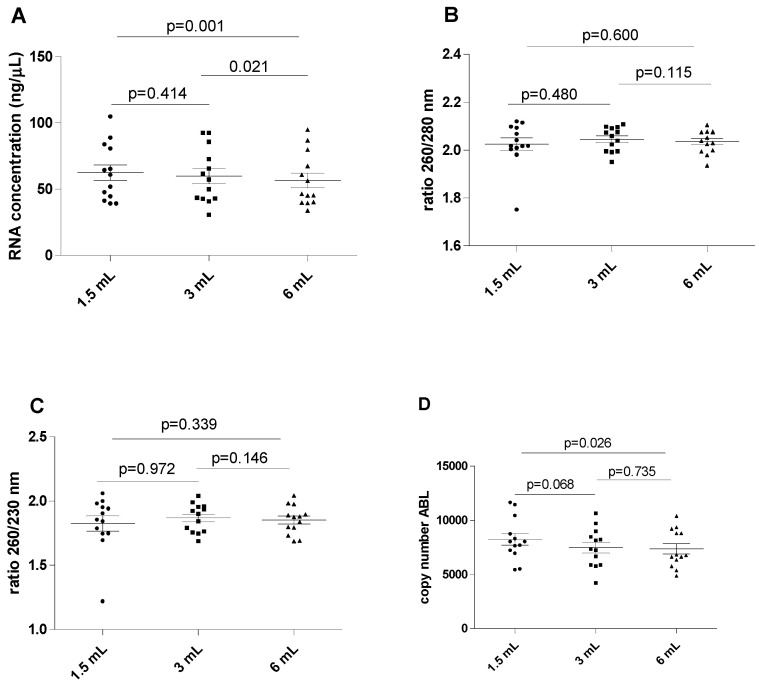
**Influence of underfilling EDTA tube on the RNA**. (**A**) RNA concentration in different blood filling tubes; (**B**) ratio between wavelength 260 and 280 nm in different blood filling tubes; (**C**) ratio between wavelength 260 and 230 nm in different blood filling tubes; (**D**) ABL gene copy number. The scatterplots shown in the panels (**E**–**H**) demonstrate that the data are paired.

**Figure 2 diagnostics-14-01934-f002:**
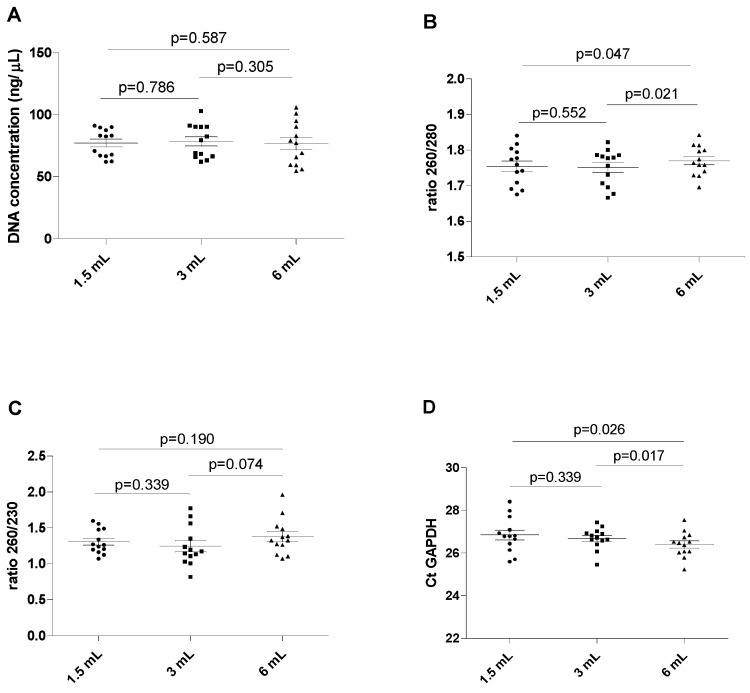
**Chart showing the assay value of the extracted DNA in different filling tubes**. (**A**) DNA concentration in different blood filling tubes; (**B**) ratio between wavelength 260 and 280 nm in different blood filling tubes; (**C**) ratio between wavelength 260 and 230 nm in different blood filling tubes; (**D**) GAPDH Ct real-time PCR. The scatterplots shown in the panels (**E**–**H**) demonstrate that the data are paired.

**Table 1 diagnostics-14-01934-t001:** Sequences of primers.

Primers	Sequences
ABL FO	TGGAGATAACACTCTAAGCATAACTAAAGGT
ABL REV	GATGTAGTTGCTTGGGACCCA
ABL PROBE	FAM-CCATTTTTGGTTTGGGCTTCACACCATT-BQ1
GAPDH FO	TCGACAGTCAGCCGCATCTTCTTT
GAPDH RE	ACCAAATCCGTTGACTCCGACCTT

## Data Availability

The data used in this article are available from the corresponding author upon reasonable request.

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
