# Peer review of "Preanalytical Impact of Incomplete K2EDTA Blood Tube Filling in Molecular Biology Testing"

_diagnostics, 2024, doi:10.3390/diagnostics14171934_

Round 1

Reviewer 1 Report

Comments and Suggestions for Authors

The manuscript entitled ‘Preanalytical impact of incomplete blood tube filling in molecular biology testing’ written by Marco Benati et al. presents interesting results regarding the evaluation of the impact of the lower amount of collected blood on the quantity and quality of nucleic acids. The condition studied in the work is particularly important due to the relatively high prevalence of incomplete blood tube filling among preanalytical errors. The presented results allow to estimate how this type of error can influence the outcome of molecular biology tests.

The manuscript is clear, scientifically sound, and presented in a well-structured manner; however, some important improvements can be introduced (see comments below). The English language is clear, but the style could be more academic.

General concept comments

    1.      The study will be more comprehensive if the used techniques will be extended for using capillary electrophoresis to assess the quality and quantity of RNA and DNA. I recommend the authors to use this method due to higher accuracy in nucleic acid quantification and, especially, the possibility of evaluating the integrity of nucleic acids by inspection of electrophoregrams and RIN calculation. The Nanodrop device used by the authors is useful to determine the contamination of DNA and RNA after isolation, but is not able to properly assess the nucleic acid integrity, which is crucial for reliability of results obtained from molecular tests.

    2.      The plots presented in the manuscript should be adjusted for pairwise analysis. They properly show differences between groups, but provide no information about whether changes are consistent across individuals. This information is very important for the proper interpretation of the obtained results (see Figure 2 in https://doi.org/10.1371/journal.pbio.1002128). I recommend the authors to add lines between paired samples (points) in Figures 1 and 2 (as in the solution for this Stackoverflow question https://stackoverflow.com/questions/67995585/plotting-paired-data-for-multiple-groups-in-ggplot).

    3.      The work focusses on differences between compared groups and evaluates their significance based on statistical significance. However, some differences are very low, despite statistical significance, and perhaps their overall significance should be questioned? For example, the authors stated in lines 150-153 “A significant difference was also found the mean of RNA concentration was compared between the 3 mL filled blood tube and the reference 6 mL filled blood tube (59.84±20.75 versus 56.58±19.94 ng/μL; p=0.021)”, but the difference is only 3.26 ng/μL and due to this reason, it maybe should not be considered significant in general? Similarly, some DNA quality ratios were pointed as significant, but the difference was very small. These issues should be addressed in the text.

I believe that my suggestions will help the authors improve the quality of their manuscript.

Author Response

  • The manuscript entitled ‘Preanalytical impact of incomplete blood tube filling in molecular biology testing’ written by Marco Benati et al. presents interesting results regarding the evaluation of the impact of the lower amount of collected blood on the quantity and quality of nucleic acids. The condition studied in the work is particularly important due to the relatively high prevalence of incomplete blood tube filling among preanalytical errors. The presented results allow to estimate how this type of error can influence the outcome of molecular biology tests.

Answer: We thank the Referee for the valuable comments on our manuscript

  • The manuscript is clear, scientifically sound, and presented in a well-structured manner; however, some important improvements can be introduced (see comments below). The English language is clear, but the style could be more academic.

Answer. We agree. As suggested, we have re-read and performed corrections

General concept comments

  • The study will be more comprehensive if the used techniques will be extended for using capillary electrophoresis to assess the quality and quantity of RNA and DNA. I recommend the authors to use this method due to higher accuracy in nucleic acid quantification and, especially, the possibility of evaluating the integrity of nucleic acids by inspection of electrophoregrams and RIN calculation. The Nanodrop device used by the authors is useful to determine the contamination of DNA and RNA after isolation, but is not able to properly assess the nucleic acid integrity, which is crucial for reliability of results obtained from molecular tests

Answer: This is a very good point, that we are aware of. Unfortunately, we didn't have this technique at our disposal. Therefore, this aspect was included at the end of the discussion as a limitation of the study, as follows: “Our study has some limitations, the first being the sample size. Further studies will be needed to explore the impact of EDTA concentration when assessing nucleic acids in larger populations. We also acknowledge that the use of capillary electrophoresis, which was not available at our institution, would allow for better accuracy in quantifying nucleic acids and establishing nucleic acid integrity by analyzing electrophoregrams and calculating RNA integrity number (RIN)”

  • The plots presented in the manuscript should be adjusted for pairwise analysis. They properly show differences between groups, but provide no information about whether changes are consistent across individuals. This information is very important for the proper interpretation of the obtained results (see Figure 2 in https://doi.org/10.1371/journal.pbio.1002128). I recommend the authors to add lines between paired samples (points) in Figures 1 and 2 (as in the solution for this Stackoverflow question https://stackoverflow.com/questions/67995585/plotting-paired-data-for-multiple-groups-in-ggplot).

Answer: Thank you for pointing this out. We add lines between paired samples (points) in Figures 1 and 2

  • The work focusses on differences between compared groups and evaluates their significance based on statistical significance. However, some differences are very low, despite statistical significance, and perhaps their overall significance should be questioned? For example, the authors stated in lines 150-153 “A significant difference was also found the mean of RNA concentration was compared between the 3 mL filled blood tube and the reference 6 mL filled blood tube (59.84±20.75 versus 56.58±19.94 ng/μL; p=0.021)”, but the difference is only 3.26 ng/μL and due to this reason, it maybe should not be considered significant in general? Similarly, some DNA quality ratios were pointed as significant, but the difference was very small. These issues should be addressed in the text.

Answer. We agree. These sentences have been added: A marginally significant difference was also found when the mean of RNA concentration was compared between the 3 mL filled blood tube and the reference 6 mL filled blood tube (59.84±20.75 versus 56.58±19.94 ng/µL; p=0.021). The differences in concentrations were globally modest, indicating that the concentration of nucleic acid may be minimally little affected by tube filling.

I believe that my suggestions will help the authors improve the quality of their manuscript.

Reviewer 2 Report

Comments and Suggestions for Authors

The authors present an original short article aiming to identify impact of laboratory tubes’ underfilling on the accuracy and quality of RNA and DNA measurement. The article is timely and interesting as preanalytical mistakes comprise a significant part of the mistakes during the laboratory diagnostics. Meanwhile, some shortcomings were revealed upon acquaintance with the article:

1.       Table 1 contains a typo in the first cell – extra text

2.       The discussion requires major changes. The first statement that I disagree with: “RNA concentrations began to decrease depending on the filling level of the blood tube” – the reference volume was 6 ml, so the fact that RNA concentration increased in case of the tube underfilling should be discussed instead. Authors explanation: “The fluctuations in RNA concentration could be due to the possible degradation of the newly released RNA, whereas DNA is almost insensitive to this aspect. Etc.” does not allow understanding of RNA increase during tube’s underfilling. The concentration of RNAses was the same in the samples compared to the concentration of RNA.

3.       “the acid nucleic quality was higher in the 1.5 mL tube than 6 mL tube for RNA, confirming the RNA concentration.” – does it mean that authors recommend underfilling to increase the quality of RNA extraction? Please, specify and provide confirmation for your point of view. In this case, though, it will contradict to the conclusion. The second part of the sentence needs to be grammatically corrected.

4.       “significant different 260/280 ratios were found for DNA in different filling tube, which could be attributable to some factors previously mentioned.” – it is not clear, how the above mentioned factors were derived in case of tube underfilling. May be the link to this article: https://pubmed.ncbi.nlm.nih.gov/15834845/#:~:text=It%20is%20demonstrated%20that%20ethylenediaminetetraacetic,tryptic%20digestion%20and%20protein%20identification could help to explain the revealed changes. If authors will provide more information, it will only benefit understanding of the obtained results.

5.       The limitations of the study should also be mentioned in discussion.

Comments on the Quality of English Language

Some sentences need to be edited. Examples:

A significant difference was also found the mean of RNA concentration was compared between the 3 mL filled blood tube and the reference 6 mL filled blood tube - …was also found WHEN (?) the mean…

but no differences were instead observed between – “instead” is wrong here

Please, carefully reread and perform corrections throughout the rest of the text as well. 

Author Response

Reviewer: 2

Comments to the Author

The authors present an original short article aiming to identify impact of laboratory tubes’ underfilling on the accuracy and quality of RNA and DNA measurement. The article is timely and interesting as preanalytical mistakes comprise a significant part of the mistakes during the laboratory diagnostics. Meanwhile, some shortcomings were revealed upon acquaintance with the article:

Answer: We thank the Referee for the valuable comments on our manuscript

  • Table 1 contains a typo in the first cell – extra text

Answer: As suggested, we have performed corrections

  • The discussion requires major changes. The first statement that I disagree with: “RNA concentrations began to decrease depending on the filling level of the blood tube” – the reference volume was 6 ml, so the fact that RNA concentration increased in case of the tube underfilling should be discussed instead. Authors explanation: “The fluctuations in RNA concentration could be due to the possible degradation of the newly released RNA, whereas DNA is almost insensitive to this aspect. Etc.” does not allow understanding of RNA increase during tube’s underfilling. The concentration of RNAses was the same in the samples compared to the concentration of RNA.

Answer: Thank you for pointing this out. These sentences have been added at the manuscript: The fluctuations in RNA concentration could be due to different activity of RNAse in the presence of high EDTA concentration. A study of Grunberg et al. (13) showed that by cap-turing Ca2+, EDTA dramatically reduces RNase I activity. In addiction Brännvall et al. (14) also showed demonstrates that Mg2+ acts in the RNase P RNA-catalyzed reaction to permit appropriate folding of RNA, thus facilitating the interaction with its RNA sub-strate, and promoting efficient and correct cleavage of newly released RNA, whereas DNA is almost insensitive to this reaction. Two pertinent references (new 13 and 14) have now been included.

  • “the acid nucleic quality was higher in the 1.5 mL tube than 6 mL tube for RNA, confirming the RNA concentration.” – does it mean that authors recommend underfilling to increase the quality of RNA extraction? Please, specify and provide confirmation for your point of view. In this case, though, it will contradict to the conclusion. The second part of the sentence needs to be grammatically corrected.

Answer. Thank you for pointing this out. These sentences have been added at the end of manuscript: Although this study confirms the importance the pre-analytic phase and the correct filling of the blood tube for molecular testing, it also demonstrates that samples would not need to be rejected when the tube is up to 75%v underfilled. This is especially important when drawing blood from patients with difficult veins, in newborns, elderly and critically ill people, in whom iatrogenic anemia should be prevented

  • “significant different 260/280 ratios were found for DNA in different filling tube, which could be attributable to some factors previously mentioned.” – it is not clear, how the above mentioned factors were derived in case of tube underfilling. May be the link to this article: https://pubmed.ncbi.nlm.nih.gov/15834845/#:~:text=It%20is%20demonstrated%20that%20ethylenediaminetetraacetic,tryptic%20digestion%20and%20protein%20identification could help to explain the revealed changes. If authors will provide more information, it will only benefit understanding of the obtained results.

Answer. Thank you for pointing this out. We added in the text:

In our study, we found no significant differences in the RNA ratio between 260/280 and 260/230, whilst significant different 260/280 ratios were found for DNA in different filled tube, which could be attributable to some factors previously mentioned, in particular to protein contamination. In the study of Janecki et al. (21), the authors found that EDTA successfully destabilizes metalloprotein structure, and this could influence the 280 ab-sorbances. Moreover, this result could be also due to the modest variations in the pH or ionic strength of the spectrophotometric solution affecting the A260/280 ratios, as earlier demonstrated by Wilfinger et al. (22). The pertinent references (now 21 and 22) were added.

  • The limitations of the study should also be mentioned in discussion.

Answer. These sentences have been added at the end of the manuscript: Our study has some limitations, the first being the sample size. Further studies will be needed to explore the impact of EDTA concentration when assessing nucleic acids in larger populations. We also acknowledge that the use of capillary electrophoresis, which was not available at our institution, would allow for better accuracy in quantifying nucleic acids and establishing nucleic acid integrity by analyzing electrophoregrams and calculating RNA integrity number (RIN).

  • Comments on the Quality of English Language

Some sentences need to be edited. Examples:

A significant difference was also found the mean of RNA concentration was compared between the 3 mL filled blood tube and the reference 6 mL filled blood tube - …was also found WHEN (?) the mean but no differences were instead observed between – “instead” is wrong here Please, carefully reread and perform corrections throughout the rest of the text as well.

Answer. We agree. As suggested, we have reread and performed all the corrections needed

Round 2

Reviewer 1 Report

Comments and Suggestions for Authors

I appreciate the efforts of the authors to respond to my comments. The lack of nucleic acid integrity assessment significantly reduces the manuscript comprehensiveness; however, I understand the authors situation. Furthermore, it was not necessary to add new panels to figures, it was enough to add lines to panels A-D making the figures space-saving with retaining all the information. However, adding lines to existing plots could be difficult due to limitations of the used statistical software. Besides, all panels on the figures should be enlarged to make them more readable. The legend of Figure 2 is covered by the figure and must be exposed. Finally, I have noticed some typo errors; therefore, I recommend the authors to read the text once more time. Due to the lack of line numbering, it was difficult to point specific places with such types of errors.

Author Response

Reviewer 1

I appreciate the efforts of the authors to respond to my comments. The lack of nucleic acid integrity assessment significantly reduces the manuscript comprehensiveness; however, I understand the authors situation. Furthermore, it was not necessary to add new panels to figures, it was enough to add lines to panels A-D making the figures space-saving with retaining all the information. However, adding lines to existing plots could be difficult due to limitations of the used statistical software. Besides, all panels on the figures should be enlarged to make them more readable. The legend of Figure 2 is covered by the figure and must be exposed. Finally, I have noticed some typo errors; therefore, I recommend the authors to read the text once more time. Due to the lack of line numbering, it was difficult to point specific places with such types of errors.

Answer: We thank the Referee for the valuable comments on our manuscript. We agree. As suggested, we have re-read and performed corrections and all panels on the figures are been enlarged
